



# Laboratory and Field Characterization of an Atmospheric Pressure Transverse Chemical Ionization Ion-Molecule reaction Region

Phil Rund [1], Ben H. Lee [1], Siddharth Iyer [2], Gordon A. Novak [3], Jake T. Vallow [4], and Joel A. Thornton [1]

[1]Department of Atmospheric and Climate Science, University of Washington, Seattle, WA, 98195, U.S.A.
[2]Aerosol Physics Laboratory, Tampere University, Tampere, Pirkanmaa, FI
[3]NOAA Chemical Sciences Lab, Boulder, CO 80305, U.S.A.
[4]University of York, Wolfson Atmospheric Chemistry Laboratories, Department of Chemistry, York, United Kingdom

**Correspondence:** Joel A. Thornton (joelt@uw.edu)

**Abstract.** We introduce a custom-built, field-deployable, atmospheric pressure Ion-Molecule reaction Region (IMR) for use with Chemical Ionization Mass Spectrometry (CIMS), the so-called "t-IMR". The design is described in quantitative detail and shows significant improvements in potential measurement interference compared to other IMR configurations, particularly those operating at low pressure. The relatively large laminar flow and inner chamber diameter reduces the probability of

sampled air and ion clusters interacting with the Teflon surfaces of the IMR before being detected by Time-of-Flight (ToF) mass spectrometry. This also leads to a substantial reduction in wall effects and artificial background signals for even low volatility organic products, exhibited by alpha-pinene ozonolysis. An electric field is induced perpendicular to flow in the t-IMR to accelerate ions and consequent charged sample clusters to the MS interface. The strength of this field is modulated and optimized to simultaneously maximize total ion flux and instrument sensitivity. A sheath flow apparatus is introduced to

provide small $N_2$ flows counter to ion and sample cluster flow into the MS to reduce the likelihood of particulate buildup and clogs to the pinhole separating the IMR from the MS, ensuring uninterrupted sampling for extended periods of time. Finally, we demonstrate the capability of the t-IMR to be deployed to the field to measure down to sub-ppt level ambient concentrations of important trace gases including reactive bromine at a ground-based site in the marine boundary layer. We find that the t-IMR design considerably reduces artificial signals from surface contact and wall effects, and improves detection of very low

concentration species in the ambient atmosphere. Future studies are recommended to evaluate the extent to which humidity effects instrument sensitivity to key compounds for the atmospheric pressure t-IMR setup.

## 1 Introduction

Multiple trace gas species present in the atmosphere at concentrations on the order of parts per trillion (ppt) and below are known to have relatively large impacts on atmospheric composition, the climate system, air quality, and human health. Chemi-

cal Ionization Mass Spectrometry (CIMS) has allowed for online and in situ measurements of many such trace gases with high temporal resolution and molecular specificity when coupled to a time-of-flight or orbitrap mass spectrometer having sufficient resolving power and mass accuracy (Du et al. 2022). Chemical ionization has the advantage of preserving the integrity of the molecules being sampled. The softer the ionization process, i.e. the lower the net energy imparted to the analyte upon ion-



ization, the less interference to analyte identification and quantification from chemical or physical processes during sampling.

Adduct ionization, where a specific ion such as iodide ($I^-$, used in all studies herein), nitrate, ammonium, etc, is used to form an ion-molecule cluster, has been shown to be especially soft compared to charge (proton) transfer, direct photoionization, or electron impact (Lee et al. 2014; Andrade et al. 2008). A key component of CIMS instruments is the Ion-Molecule reaction Region (IMR), where neutral analyte molecules obtain a charge (positive or negative) via charge transfer, reactive charge transfer, or ion-neutral adduct formation. The resulting analyte ions or clusters can then be guided by electric fields and fluid dynamics

inside the mass spectrometer (MS).

The IMR design significantly impacts the sensitivity and selectivity of a CIMS to certain classes of compounds and/or total ion flux, both of which determine the limits of detection (LOD) at which atmospheric components may be measured (Robinson et al. 2022; Liu et al. 2019; Lopez-Hilfiker et al. 2016; Slusher et al. 2004; Huey et al. 1995). Many previous IMR designs involving the $I^-$ reagent ion utilize a low pressure chamber ($< 0.1$ atm). The low pressure has practical and analytical benefits,

such as allowing larger apertures for ions to enter the MS, reduced ion-molecule interaction times to minimize secondary ion chemistry without needing electric fields or high flows, and lower absolute water vapor concentrations, as humidity-dependent sensitivities have been reported for multiple compounds across CIMS measurements (Krechmer et al. 2018; Zhao et al. 2017; Kercher et al. 2009; Dörich et al. 2021). However, by reducing the pressure of the IMR dilutes sample molecular number concentration by up to several orders of magnitude (Rissanen et al. 2019). This leads to a lower collision frequency between

analytes and reagent ions for a given reaction time and thus lower ultimate sensitivity. In a low pressure IMR, it has been shown that even a relatively small increase in operating pressure can increase sensitivity (Novak et al. 2020). For already low concentration species, these issues could prevent in-situ measurements of sub-ppt level compounds (Jokinen et al. 2012). Moreover, the expansion upon sampling through an orifice from approximately 1 atm to $<0.1$ atm coupled with physical constraints to keep interaction timescales short, leads to turbulent mixing and enhanced wall interactions.

When considering IMR design, the two main objectives to consider include first generating a sufficient and stable stream of reagent ions to produce signal for analyte species. And of equal importance, the verification that such signals are a true representation of the gaseous sample, ambient or otherwise. This involves disturbing the sample gas as little as possible before it enters the CIMS. Multiple challenges arise in the pursuit of achieving this goal while maintaining a sufficient number of ion collisions for continuous measurement. One such problem is chemical reactions occurring in and on the surfaces of the

IMR, which would alter the molecules before detection, thereby creating an artificial chemical signal not present in the actual atmosphere being sampled. One way to reduce this effect is to move the sample through the IMR as quickly as possible to reduce the likelihood of interaction with other molecules and/or surfaces, while ensuring sufficient ion intrusion. The residence time, or average time that an air mass spends in the IMR chamber before entering the CIMS, is controlled mostly by the pumping speed from an external air pump. Another consideration beyond flow speed is turbulent mixing, which is dependent

on the pumping scheme and angle at which air enters the CIMS. Turbulence can decrease the likelihood of ion collisions and interfere with total detected signal, if not eliminate signals completely. Designing an IMR such that the inlet flows are as laminar as possible helps to eliminate small eddy formations and turbulence which increase molecular interactions and disturb reagent flows (Yang et al. 2024). There are also so-called IMR "wall effects", which refer to a situation where ambient molecules are





deposited onto or absorbed into the surface of the IMR to potentially be evaporated and reintroduced into sample flow at a
later time. This not only causes an artificial depletion in a signal at the time of sampling, but also can create an enhancement
in detection of that particular compound later, both of which misinterpret current ambient conditions and interfere with precise
time of day measurements required to understand trace gas chemistry (Lopez-Hilfiker et al. 2014). Advancements in IMR
design have been researched in pursuit of rectifying this commonly observed effect (Palm et al. 2019; Crounse et al. 2006).
Choice of IMR surface material and experimental verification of wall effects are thus crucial to the reliability of resultant
chemical signals.

This work introduces and demonstrates the capabilities of a field-deployable atmospheric-pressure "transverse IMR" using
iodide reagent ions generated using a vacuum ultraviolet (VUV) lamp. This updated design is based off of a previous transverse
IMR setup from Zhao et al. (2017) constructed with the intent of minimizing these common problems including turbulent
mixing and wall interactions which lead to artificial and/or depleted CIMS signals. We describe in detail its design and evaluate
performance based both on metrics determined in laboratory experiments as well as ambient measurements in the field.

## 2 t-IMR Description

We describe a Chemical Ionization inlet coupled to a Time of Flight Mass Spectrometer (i.e. a ToF-CIMS). We utilize a
Tofwerk AG high resolution ToF-MS (model L-ToF). The design and operation of the ToF-MS and general coupling to chemical
ionization regions has been described previously (Bertram et al. 2011; Lee et al. 2014; Lopez-Hilfiker et al. 2014; Riva et al.
2019). Briefly: analyte ions and remaining reagent ions are guided through a series of differentially pumped vacuum chambers,
by segmented quadrupoles, one in the first stage downstream of the IMR with typical operating pressure of 2 mbar, and the
other in the second pumping stage with pressures on the order of 1e-3 mbar. Ions are then extracted into the ToF region
(pressures typically on the order of 1e-7 mbar) where they receive a pulse of energy and fly along a U-shaped path to a Micro
Channel Plate (MCP) detector. In this process, the sample molecules are separated by mass due to differences in flight time.
From these flight times, exact masses are calculated and binned resulting in the spectrum of ion counts versus mass-to-charge
ratios for further analysis. For all data presented in this study, the default timescale of collected spectra is 1 Hz. Integration of
peaks in each mass spectrum results in timeseries of all identified species, reported in default units of counts from the MCP
in the ToF region. In order to convert detector counts to the relevant atmospheric concentration, a sensitivity value must be
determined for every identified species (Lopez-Hilfiker et al. 2016). Sensitivities are typically quantified in units of counts/ppt
and are preferably determined by direct calibration, i.e. introducing a known concentration of a given species into the IMR and
recording the corresponding instrument response for a set period of time such that the signal is stable and reproducible.

Shown in Figure 1 is a schematic of our new transverse IMR, henceforth referred to as the "t-IMR", where the inlet is
comprised of a 1 inch (2.54 cm) outer-diameter (OD) Teflon™ tube with 1/16 inch (0.16 cm) wall thickness housed in an 20.3
W x 7.6 H x 3.8 D cm aluminum block, which is secured to the ToF-MS. The distance from the upstream edge of the 1" OD
tube (i.e. the entrance to the IMR region) to the reagent intersection/MS ports at the center of the aluminum block housing is
32.4 cm, and the total length of the IMR tube is approximately 47 cm. The aluminum block is used mainly for stability, port



alignment, and ability to be electrically grounded. Thus, when mentioning to the t-IMR in this work, we are referring to the 1" OD teflon inlet tube acting as the ion-molecule reaction chamber, which is noted above as longer than the block housing it. This design brings ambient air into the ionization region continuously such that the sampled air flow does not experience any

steps in diameter, corners, or other wall materials until it is exhausted beyond the aluminum block and capillary MS entrance area. The 1" OD Teflon tube extends upstream of the housing to ensure development of laminar flow and sampling outside of the overall instrument boundary layer. Figure 1 highlights the laminar sample flow of 10 L/min (blue arrow) intersected by reagent ions (red arrow) which are accelerated by the induced electric field created by the applied potential between the ion source exit and the grounded capillary entrance to the ToF-MS, where newly formed analyte ions enter the MS, indicated

by the purple arrow. Iodide reagent ions are generated via a gaseous mixture of methyl iodide ($CH_3I$) and toluene ($C_6H_5CH_3$) carried by ultra-high purity nitrogen gas (UHP $N_2 \geq 99.999\%$) at a combined standard flow rate of 1.5 liters per minute (LPM). This gas mixture is then introduced into the cavity of an ionization device, ultimately producing $I^-$ reagent ions introduced into the t-IMR at the point of the applied electric potential, indicated in orange in Figure 1. A previous version of the atmospheric pressure, transverse IMR without the VUV ionization source is described in work by Lee and coauthors examining pinene

oxidation (Lee et al. 2023).

Shown in the gold-bordered inset of figure 1 is a cross-sectional view of the t-IMR, where the sample flow direction points out of the page, reagent ion flow is shown in red, the sample + ion flow into the CIMS in purple, and electric potential application indicated with orange dots (with ground in black), analogous to the rest of the schematic. The critical orifice used to separate the IMR from the small segmented quadrupole (SSQ) region (the MS entrance) is a small cone that is 10 mm in

length which extends approximately 2.5 mm into the t-IMR chamber, and has a pinhole of diameter of 0.20 mm where air enters the capillary. The internal cavity downstream side of the orifice is also conically shaped, the diameter expanding to 3.7 mm toward the SSQ region from the pinhole interface. The orifice, highlighted in purple in the inset, is designed specifically to maintain the pressure difference between the IMR at a somewhat dynamic atmospheric pressure on the order of ∼1000 mbar and the SSQ operating at 2 mbar. Given the small size of this capillary separating the CIMS from the IMR, buildup and/or

clogging due to particulates can be a concern depending on operating conditions. This is addressed via the addition of a sheath flow apparatus implemented to provide a small concentric flow of UHP nitrogen gas around the capillary, running opposite in direction to the flow being sampled through it into the MS (shown with green arrows in the inset). This small counter-flow prevents a significant fraction of large particles from depositing onto the capillary which could lead to eventual blocking of the sample orifice. The sheath flow therefore allows for prolonged sampling periods (at least several days) uninterrupted by

cleaning that would be otherwise necessary to clear blockages and prevent an interruption in the flow between the IMR and the SSQ chamber. Additionally, this sheath flow likely acts to partially dry the ion flow into the MS, leading to the evaporation of weakly bound water clusters formed when sampling high humidity air.

A diagram of the ionization setup used in all studies presented in this work, is shown in figure A1 . Both the block and inlet tube contain ports for reagent ion flow generated with a VUV photo-ionization device (Jiang et al. 2023), as well as a port

for the critical orifice or capillary entrance between the SSQ region of the CIMS and the atmosphere/t-IMR. Use of VUV to generate iodide ions for CIMS work has been demonstrated previously by Ji et al, who report similar instrument sensitivities,



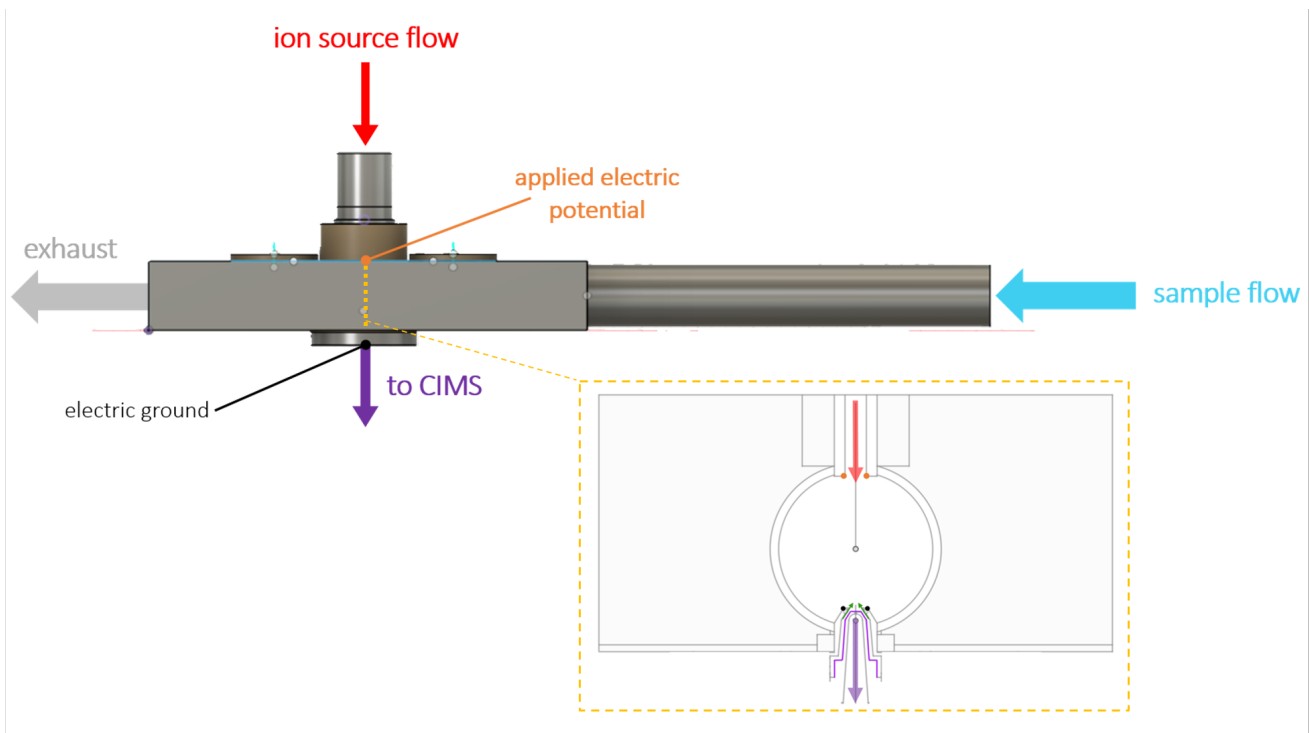

**Figure 1.** Schematic of the transverse IMR (t-IMR). A continuous 1" OD Teflon inlet held in place by stainless housing carries sample air at atmospheric pressure at a flow rate of 10 L/min where it is then intersected at 90° by a 1.5 L/min flow carrying the $I^-$ reagent produced from a VUV ion source, which is accelerated by an electric potential toward the CIMS. The gold dashes in the housing schematic and corresponding inset show the location and a detailed cross section of the t-IMR. Reagent flow is again shown in red while sample + ion flow into the CIMS is shown in purple. For the inset the standard 10 L/min flow through the t-IMR cavity (ID = 7/8") points out of the page. The critical orifice conical capillary is highlighted with purple outlining. The smaller sheath counter-flow is indicated with green arrows. Electric potential application is shown with orange points, with electric ground in black.

reagent ion production, and resultant mass spectra to those obtained with established radioactive sources (Ji et al. 2020). Our VUV ionizer setup uses a Heraeus PKS106 UV lamp held in custom stainless steel housing, the base of which is the cavity for ionization, including a female Swage fitting for 1/8" OD tubing (reagent flow input for this study) along with a 1/4" OD output tube (ion output) built in. The 1/8" port acts as the entrance for both the reagent and carrier gases, which here are the $CH_3I$ + toluene and UHP $N_2$, respectively. The UV lamp utilizes krypton gas to generate UV ionization light with a photon energy of 10.6 eV while requiring a DC voltage of 1.5 kV. Once in the lamp housing cavity, $CH_3I$ is ionized by the UV radiation supplied by the lamp, aided by toluene acting as a photosensitizer. Optimal flows for instrument operation were determined to be 1.5 LPM through the $CH_3I$ permeation tube channel and 2 standard cubic centimeters per minute (sccm) over the head space of a vial containing a volume of 2-3 mL of toluene.



The VUV setup produces an outflow of I$^-$ ions carried in the UHP N$_2$ from the VUV housing into the t-IMR chamber. The use of the VUV marks a notable difference between this design and that of Zhao et al (2017), as it does not require the use of salt solutions for ion production as the extractive electrospray ionization (EESI) method does, but rather uses a slow release permeation tube. This prolongs the potential continuous sampling of the t-IMR CIMS, as salt solutions need not be replenished

for reagent production, while also avoiding potentially clogging of the capillary delivery tubes involved with EESI. Both theirs and our setups eliminate the need for radioactive sources used in many previous CIMS experiments, which can be difficult to deploy to remote field sites, as well as presenting a higher potential hazard level for operation. Though it is important to note that our VUV device may be replaced by any other ionization instrument (including a radioactive source if so desired) at the user's discretion.

One notable effect of the specific VUV orientation in this setup is the exposure of sample air to a small amount of UV radiation, which has the potential to change sample composition as well as create O$_2$$^-$ ions which can create unintended additional analyte ions. These O$_2$$^-$ cluster peaks may appear at similar mass-to-charge ratios (m/Qs) and potentially complicate the identification of certain compounds clustered with I$^-$ in the negative mode for the CIMS. This has been reported by Ji et al. (2020) and is diagnosed in our setup mainly by the signal of the fitted compound CO$_3$$^-$ at m/Q 60 in the mass spectrum. In all

laboratory experiments presented here, it is found that the mean CO$_3$$^-$ signal is always less than 0.5% of the reagent signal (I$^-$ + IH$_2$O$^-$), in many cases under 0.05%, and does not interfere with the identification and quantification compounds of interest. This is due to the much higher signals for iodide clusters and importantly the mass defect of I$^-$ (separated from O$_2$$^-$ clusters) easily identified with the high resolution LToF instrument. That said, it is recommended for generally cleaner spectra to reduce sample exposure to the VUV lamp, as done by Ji et al with a 90 degree bend in their reagent delivery line, which is a design

element we will likely incorporate into future versions. High CO$_3$$^-$ signals are typically observed when air leaks are present in the UHP N$_2$ reagent delivery lines, as even a minuscule amount of air can lead to an O$_2$ concentration similar to that of CH$_3$I in the UV-illuminated volume, which would likely result in O$_2$$^-$ clusters present in spectra.

The t-IMR chamber is sealed against the ionizer and SSQ entrance via, press-fit O-rings and screws holding the block to the CIMS face, such that the t-IMR is only open to ambient air at it's entrance and ultimately the external pump throttle to control

the flow rate. The ionization and critical orifice ports are aligned for reagent flow to be perpendicular to and combine with sample flow. The standard flow rate chosen for operation of the t-IMR is 10 L/min in an effort to minimize residence time of sample molecules in the IMR while maximizing total ion counts (TIC) detected by the CIMS, as well as maintaining laminar flow. Laminar flow is achieved given the Reynolds number (Re) calculated by treating the t-IMR as a pipe:

$$Re = \frac{4\rho_{air}V}{\pi\eta d} \tag{1}$$

where $\rho_{air}$ is the density of air (kg m$^{-3}$), $V$ is the volumetric flow through a pipe (m$^3$ s$^{-1}$), $\eta$ is the viscosity of air (Pa s), and $d$ is the diameter of the IMR tube (m). The calculation at the ambient temperature of 293 K yields a Reynolds number of Re = 634. Flow through a pipe is defined to be laminar if the Reynolds number is determined to be less than 2000 and is considered turbulent if greater than 4000 (Hinds 1999). So for our standard flow of 10 LPM and lower flows, the t-IMR operates within



a laminar flow regime. Given this flow rate and the distance from the sample inlet to the CIMS port, the calculated average residence time of sampled molecules is 0.75 seconds.

Reagent ions are introduced orthogonal to the sample flow via both a relatively small carrier flow and a DC electric field which accelerates the ions across the higher atmospheric pressure cross-flow to the pinhole, similar to the previous design. This electric field is achieved with an electric potential bias provided via connecting an external adjustable voltage power supply to the metallic (conductive) exit of the VUV ionizer cavity (see fig A1) and electrically grounding the capillary dividing the CIMS SSQ region from the t-IMR. The subsequent electric force accelerates reagent ions directly across the IMR tube (perpendicular to the flow, see below), colliding and combining with sample molecules, and continuing to the grounded capillary, after which the charged molecules are guided by the standard CIMS octopoles and other internal electronics. The strength of the applied electric field determines ion-molecule interaction time, the total ion current which reaches the capillary/MS, as well as the ion energy and thus the ion-analyte cluster stability. The effect of varying this applied potential and resultant electric field in the t-IMR is presented in this work and, unless otherwise specified, the standard voltage applied to the exit point of the ionizer is 1.5 kV. Ion-molecule interaction time is determined as the time for $I^-$ ions to travel across the t-IMR chamber and calculated by using a simplified electric field strength calculated as $E = \frac{V}{d}$, the applied voltage divided by the distance across the t-IMR chamber ($d$), i.e. the inner diameter (ID) of the IMR tube, 22.225 mm. Combining the electric field strength with the ion mobility of $I^-$ determined in a previous study to be between 2.15 - 2.49 $cm^2$ $V^{-1}$ $s^{-1}$ when clustered with 0-2 water molecules (Wolańska et al. 2023), the resulting ion-molecule interaction time is 1.3-1.5 ms. In order to prevent buildup of static charge or induced electric fields on different surfaces, grounding wires are also fixed to the t-IMR metal housing as well as the sheath flow apparatus input. The use of conductive metal (compared with full Teflon block) as IMR housing, as well as a sheath flow apparatus for the capillary entrance to the CIMS, both of which held at electric ground, demonstrate notable improvements upon the previous crossflow design. These new additions create conditions for more stable instrument signal, longer uninterrupted sampling for ambient measurements, and lower potential electrical interference i.e. from charge buildup. The VUV with applied potential and capillary setup also allow for the CIMS instrument to sample in any orientation.

Background signal determination for CIMS measurements is crucial for accurate signal reporting. This involves a period where sample gases are prevented from entering the CIMS while keeping IMR pressures constant, which results in mass spectra that reflect the so-called background signals for chemical components intrinsic to the CIMS at a given time. These background signals can shift over time for any detected mass, so acquiring background spectra as frequently as possible is important. In this case, the background determination or "zero" signal is acquired by overflowing the t-IMR with UHP nitrogen, a method used in other CIMS studies (Lee et al. 2018; Palm et al. 2019). Here, between 12 and 14 LPM of nitrogen is introduced to the 7/8" ID IMR chamber, acting as an "overflow" of the standard 10 LPM sample flow. The nitrogen delivery apparatus is moved directly in front of the center of the t-IMR inlet and removed when returning to regular sampling. The large amount of UHP N2 in this configuration is not always practical depending on the sampling location. As we show below, the relatively low wall interactions enable less frequent background determinations. Moreover, alternative background determinations using scrubbers could also be employed instead of UHP N2. The spectra and subsequent signals recorded during a zero period are used in background determination, as well as limit of detection calculations (Bertram et al. 2011). Low pressures in the BSQ





and ToF chambers of the CIMS are maintained by a Pfeiffer Splitflow turbo pump, which in addition to the SSQ are back by
an Ebara model EV-PA 500 pump. The Ebara pump is throttled with a manual valve to achieve a pressure of 2 mbar when the
critical orifice is open to ambient pressure.

## 3 Sensitivity and Transverse Electric Field Effects

The ability to set an applied voltage to create an electric field propelling ions generated by the VUV lamp across the t-IMR
cavity is critically important to producing signal with the CIMS, and allows the use of such a large sample flow (10 L/min).
Without any applied field, total ion counts remain near zero, presumably from physical loss due to ions being forced into
the exhaust line by the high t-IMR flow. This result also suggests neutral components of the reagent ion carrier flow arising
from the ionization process or precursor gases do not get entrained into the main portion of the sample flow that interacts
with the reagent ions. However, even with a small field applied it is apparent that electric forces begin to compensate this
kinetic force. It is therefore paramount to characterize the optimal electric field strength in normal operation to maximize
signal and sensitivity. The standard electric potential of 1.5 kV applied at the VUV ionizer exit is chosen based on experiments
performed to evaluate changes in ion signals and sensitivity while varying this voltage. The experimental setup includes the
t-IMR sampling laboratory room air at the standard 10 L/min flow rate, along with an added 0.040 L/min flow of bromine gas
($Br_2$) produced by a KINTEK™commercially purchased $Br_2$ permeation device, carried by UHP $N_2$ gas. The permeation tube
is kept under constant flow at a temperature of 40°C regulated by heating tape and a thermocouple with digitally monitored
temperature readings. The published output of the device held at this temperature is 140 ng/min. The permeation rate is also
measured gravimetrically, independently confirming this published rate within 3%. Using this value and the measured flow
rates of the carrier $N_2$ gas and t-IMR, the resultant concentration of $Br_2$ (in ppt) present in the IMR chamber during this
constant addition is calculated. Using measured counts from the CIMS detector for the $Br_2I^-$ cluster peak, normalized to Total
Reagent Ion Counts (TRIC = $I^-$ + $H_2OI^-$) with this known concentration, the sensitivity value for bromine is calculated and
tracked throughout the experiment.

When considering the theoretical effect of the applied electric potential, the goal is to have the resultant electric field be
sufficiently strong to accelerate enough generated $I^-$ ions across the high volume flow passing through the t-IMR to make it to
the capillary CIMS entrance. The stronger this field, the higher the kinetic energy of these ions, which increases the theoretical
probability of $I^-$ ion and cluster transmission to the CIMS and therefore total ion signal. However, maximizing kinetic energy
is not the singular goal as there is also in theory a point at which the velocity of the lone $I^-$ ions becomes so large that the
likelihood of them attaching to analyte molecules in the sample flow begins to decline. The result of such a high electric field
would in theory be higher TRIC, but lower signal counts for the compounds of interest which would result in lower instrument
sensitivity; along with the safety and power consumption concerns that come with indefinitely increasing the electric potential.

Shown in Figure 2 is the result of this experiment, showing the relationships of both $Br_2$ sensitivity in counts $ppt^{-1}$ per
million TRIC with the applied electric potential measured in volts (V) on the left, along with TRIC and total ion counts (TIC)
on the right. As the applied voltage is increased, the measured CIMS sensitivity to bromine decreases, while both TRIC and





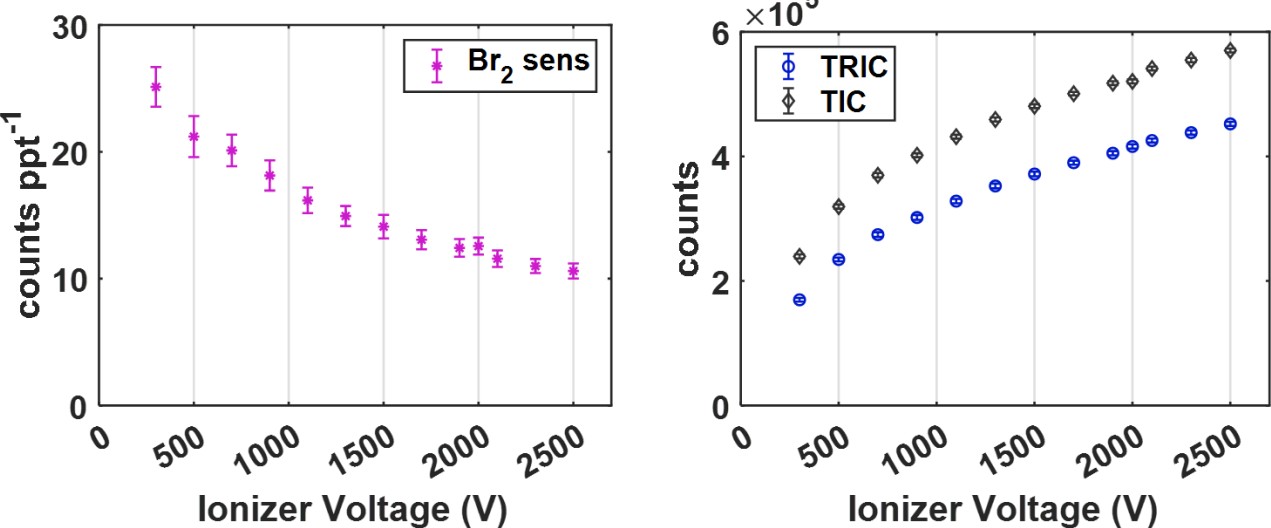

**Figure 2.** (left) Br$_2$ average sensitivity (counts ppt$^{-1}$ per 1e6 TRIC) vs. electric potential applied to VUV ionizer and (right) average Total Reagent Ion Counts (TRIC = I$^-$ + IH$_2$O$^-$ signals) and Total Ion Counts (TIC) vs. ionizer voltage.

TIC increase. The changes in both are non-linear with electric field strength, showing the larger changes at voltages below ~1500 V. Br$_2$ sensitivity shows a slight exponential decay with applied voltage, decreasing by more than a factor 2 between 300 and 2500 V. TRIC conversely shows a large increase which tapers off at higher voltages. The difference between TIC and

TRIC shows a similar increasing pattern, indicating that all other analyte signals present in room air plus that of the bromine added increase as well. Ultimately, the 1.5 kV setting is chosen for t-IMR operation as a compromise between to goal of maximizing both instrument sensitivity as well as total ion counts.

## 4 Wall Effects

Another improvement with the transverse design of the t-IMR is the reduction of so-called wall effects. As mentioned in

the introduction, wall or memory effects refer to compounds depositing onto IMR surfaces leading to both lower signals for species at the actual time of sampling as well as increased artificial signal later when molecules are liberated from surfaces and reintroduced in the sample flow. Reducing artificial signal is important for all CIMS applications, particularly for those wanting to measure low concentration species. Multiple laboratory studies have been conducted to quantify and reduce this unwanted effect across different IMRs and instrument techniques. Palm et al. (2019) showed that the fast zero technique in their new low

pressure IMR significantly reduces memory effects quantified as the "instrument delay time", which is defined as the time it takes for a specific compound's measured signal to return to 10% of its value during a controlled addition to the IMR. In the same study, Palm and coauthors also provide a summary of how instrument delay time generally decreases with the calculated saturation vapor pressures and subsequent volatilities (or C* values) of various compounds tested. In other words, less volatile





species (lower C*) generally take longer to evaporate off of IMR surfaces after deposition and will contribute to artificial signal
for longer periods of time (longer instrument delay time). The lowest volatility species presented in this study had instrument
delay times on the order of 10 minutes or longer. The goal for the t-IMR is to minimize this effect as much as possible by
reducing molecule-surface interactions in the instrument.

To characterize our t-IMR in regard to wall effects, experiments were conducted to generate a suite of organic molecules
covering a large range in volatility, via alpha pinene ozonolysis. The setup for this experiment is shown in figure A2. Ozone
was generated by flowing approximately 4 LPM of zero air from a Teledyne model 701 zero air generator through an enclosed
cavity containing an ultraviolet lamp with output at 254 nm wavelength (Jelight #82-3309-9). Ozone concentrations registered
between 900-1000 ppb in a glass flow tube where it was mixed with a 200 sccm flow of alpha pinene + UHP nitrogen gas. This
resulted in a suite of organic products detected by the iodide CIMS, including various levels of oxygenated pinene products (i.e.
$C_{10}H_{16}O_x$). The products were introduced to the t-IMR by manually moving a 1/2" OD teflon tube outlet from the glass flow
tube in front of either the t-IMR entrance or to exhaust. This output was moved in front of the t-IMR for 5, 10, and 30 second
pulses repeatedly throughout the experiment to determine if sampling time influences instrument delay on these timescales.

Shown in figure 3 are the 1 Hz timeseries of thirty five alpha pinene ozonolysis products from this experiment during a rep-
resentative 30 second pulse. Relative signals for each detected compound, normalized to their mean value during the 30 second
pulse are shown colored by their oxygen-to-carbon atom ratios (O:C), indicating the least oxidized in green (O:C < 0.5) to most
oxidized in purple (O:C > 1.5). Also plotted with red dashed lines is the value 0.1, representing the threshold corresponding to
10% of the mean signal for each compound. The absolute signals measured corresponding to concentrations) for these products
are consistent across the multiple additions and varied times, reaching the same level with each pulse. The signals drop to be-
low their respective thresholds (red dashed line) on the order of 1-2 seconds after the products outflow tube was removed from
the t-IMR, at 22:35:51 (formatted as UTC hh:mm:ss) as illustrated by the bottom plot of figure 3 showing a zoomed in view
of the end of the 30 second pulse (highlighted in the grey dashed box for the top plot). Two representative organic molecular
compositions are included in this plot: $C_{10}H_{16}O_3$, likely a collection of relatively high concentration semi-volatile products,
and the much less volatile, more oxygenated products having a compoisition of $C_{10}H_{16}O_8$. These representative products of
$\alpha$-pinene ozonolysis span multiple orders of magnitude in volatility space. C* values were calculated at temperatures of 298 K
for these two compounds using their saturation vapor pressures as calculated and published by Hyttinen et al. (2022) using the
SIMPOL methodology outlined by Pankow and Asher (2008). The C* value determined for $C_{10}H_{16}O_3$ is $1.04 \cdot 10^3 \frac{\mu g}{m^3}$ and that
of $C_{10}H_{16}O_8$ is $26 \frac{\mu g}{m^3}$ as an upper limit, and $1.7 \cdot 10^{-4} \frac{\mu g}{m^3}$ as a lower limit. This experiment demonstrates that over a wide range
of chemical volatility (and molar mass), instrument delay times remain between 1-2 seconds, consistent with inlet residence
time. Contextualizing these results with the summary of instrument delay time vs. saturation vapor pressure from Palm et al,
the atmospheric pressure t-IMR demonstrates significant improvement in reduction of wall memory effect for lower volatility
organic compounds, especially after accounting for the t-IMR residence time of 0.75 s. The t-IMR instrument delay time on
the order of 1 second is much less than the 10-minutes reported for low pressure designs. This result is also comparable if not
an improvement to the e-folding time (time for signal to drop to 36% signal) of 1s calculated for the crossflow design presented
by Zhao et al. (2017) for nitric acid.



**Figure 3.** 1 Hz timeseries of 35 $\alpha$-pinene ozonolysis products generated in a laboratory study and detected in the t-IMR. Compounds shown colored by O:C ratio and reported in normalized signals relative to the averaged detected during addition. Relative 10% signal lines shown in red. Bottom plot shows zoomed in view of timeseries as shown in grey box of top plot.





## 5 In-Situ Observations

The t-IMR was deployed to the field as part of the Bermuda boundary Layer Experiment on the Atmospheric Chemistry of Halogens (BLEACH). The BLEACH campaign took place over two six-week periods from May-June 2022 and January-February 2023. The results presented here will focus on the latter winter leg of the measurements. The t-IMR CIMS was stationed at the Tudor Hill Marine Atmospheric Observatory (THMAO) located on the southwestern shores of Bermuda operated by the Bermuda Institute of Ocean Sciences (BIOS). The CIMS sampled from atop a 10 meter tower on the shore and

was contained in weatherproof housing with an attached air conditioner to assist in stabilizing instrument temperatures. To both accelerate ambient air to the IMR chamber as well as shield the inlet from precipitation and high winds, a 3" diameter secondary inlet was sealed around the 1" t-IMR chamber and was pumped at 110 L/min, of which 10 L/min was sub-sampled by the t-IMR with the same setup of 1.5 kV applied electric potential and iodide ionization scheme described in the previous sections. The additional residence time from this 3" secondary inlet at such a pumping speed is calculated to be 0.76 s. which

brings the total residence time for ambient molecules to 1.5 s from entering the outer inlet to the pinhole CIMS entrance. Background signals for a given compound were determined by taking the average signal for that compound during a "zero period" where a flow of 14 L/min of the UHP $N_2$ gas overflowed the t-IMR by way of a mass flow controller output manually placed in front of the 1" inlet (within the larger 3" sheath inlet). These signals are characterized by a local minimum in the $H_2OI^-$ signal, as the ionization region is being overflowed with virtually completely dry nitrogen gas as opposed to the more

humid ambient conditions. This "zero" signal is then averaged between multiple overflow periods throughout the campaign and subtracted from the corresponding ambient counts for that detected species. LOD is also calculated using signals recorded during zero periods according to the equation from Bertram et al, giving detection limits on the order of tens of ppq for certain low concentration halogenated species, including HOBr. The t-IMR CIMS in iodide mode is capable of measuring multiple reactive halogen species as well as a suite of organic molecules in the ambient conditions at the site.

Shown in Figure 4 is the timeseries as well as the diurnal profile for ambient HOBr measured during the winter BLEACH campaign. Median concentrations are shown in the open circle trace for the diel cycle, with shading representing the $25^{th}$ and $75^{th}$ percentiles for the entire sampling period. This data is the result of peak-fitting and integrating 1-minute pre-averaged spectra from the t-IMR CIMS, after which further binning is performed to give a 15-minute data frequency. The timeseries is quality controlled to exclude any calibrations, zeros, or instrument maintenance. Quantum model simulations to calculate

the binding enthalpies (BE) to the reagent $I^-$ for each of these molecules were performed, as these values correlate with and therefore may be used as a proxy for the relative sensitivities between these species (Iyer et al. 2016). Geometries of the free molecules and the molecular clusters are optimized at the PBEPBE/aug-cc-pVTZ-PP level of theory using the Gaussian 16 program (Perdew et al. 1996; Kendall et al. 1992; Frisch et al. 2016). Iodine and bromine pseudopotential definitions are taken from the Environmental Molecular Sciences Laboratory (EMSL) basis set library. The final energies are refined at the

DLPNO-CCSD(T)/def2-QZVPP level of theory using the ORCA program version 4.2.1 (Feller 1996; Peterson et al. 2003; Riplinger et al. 2013; Riplinger and Neese 2013; Weigend and Ahlrichs 2005; Neese 2012). The calculated BEs are 24.3 and 15.1 kcal/mol respectively for $Br_2$ and HOBr. The regular calibrations for $Br_2$ throughout the campaign are used to produce





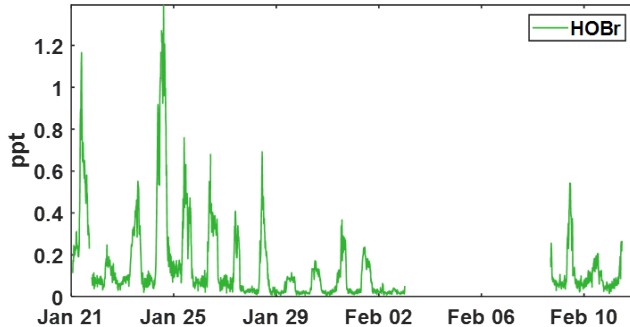 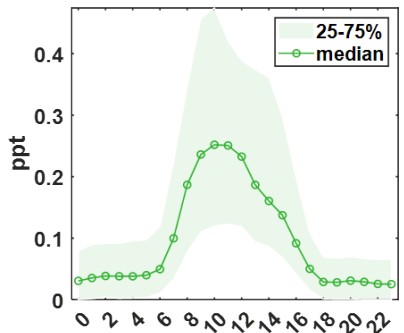

**Figure 4.** (Left) 15-minute averaged timeseries of ambient HOBr concentrations measured during the winter BLEACH campaign and (right) the diurnal profile showing median signal at each hour of day (HOD) in local time. Shading indicates the bounds of the $25^{th}$ and $75^{th}$ quartiles for the measured concentrations in each hourly bin.

a general sensitivity value for the t-IMR CIMS, which is used for the conversion of instrument signal counts to atmospheric concentration (ppt). Using the binding enthalpy relative to that of $Br_2$, we scale the measured $Br_2$ sensitivity value and convert
measured signals of HOBr to atmospheric concentration as well:

$$\frac{\log C_{f,HOBr}}{\log C_{f,Br2}} = \frac{\Delta H_{HOBr}}{\Delta H_{Br2}} \Rightarrow C_{f,HOBr} = C_{f,Br2}^{\left(\frac{\Delta H_{HOBr}}{\Delta H_{Br2}}\right)} \qquad (2)$$

Where $C_{f,X}$ is the instrument sensitivity and $\Delta H_X$ is the binding enthalpy for measured compound X. Using equation 2, BE, and the in-field calibrations, the calculated sensitivities for $Br_2$ and HOBr are respectively 1.5 and 1.3 counts ppt$^{-1}$ per 1e6 TRIC. Instrument signal counts (from integrating peaks in spectrum) are also normalized to 1e6 TRIC, giving results in
atmospheric concentration (ppt) after applying sensitivity values. The daily maximum levels of HOBr vary throughout the campaign, but concentrations reliably increase during daylight hours roughly between 7:00 and 17:00 hours during the winter in Bermuda, corresponding to within the hours of sunrise and sunset for THMAO in Jan-Feb. This aligns with the current understanding of the bromine chemical mechanism in that most gas-phase production of HOBr in the atmosphere requires photochemical reactions (Zhang et al. 2023; Swanson et al. 2022).

335       Shown in figure 5 is the timeseries and diel cycle for nitric acid concentrations (HNO$_3$) as measured by the t-IMR CIMS. Similarly to those of the HOBr data, 1-minute averaged spectra were integrated to produce signal counts. However, in this case, the signals are converted to atmospheric concentrations using a direct calibration of the t-IMR after the campaign. The calibration utilizes a custom made permeation device where concentrated nitric acid fills a permeable PTFE tube sealed at each end with solid PTFE caps. The output of this permeation device is determined gravimetrically and converted to units of
ng/min, which is then used in conjunction with total IMR flow to determine the concentration in the t-IMR and subsequently the sensitivity to HNO$_3$ in the same manner as the direct calibrations of $Br_2$ described earlier. What was determined to be a small contamination of nitric acid on tubing connections involved during zero periods in the BLEACH campaign made it such that the HNO$_3$ timeseries could not be background subtracted, though all other quality controls used in the HOBr data





are applied here. The diurnal cycle shows median concentrations increasing during daytime hours evident in the shading of
the 75$^{th}$ percentile measurements, with a particular period of higher median values in the morning. The relatively elevated
concentrations during the day are consistent with the main gas phase production of HNO$_3$ coming from reaction of NO$_2$ with
OH, requiring photochemistry. This is evidenced by the bottom two plots shown in figure 5 which show a zoomed in view
of a peak in HNO$_3$ (cutoff by axis limits in the complete timeseries for clarity) accompanied by NO and NO$_2$ concentrations
on the bottom plot as measured by a Laser-Induced Fluorescence (LIF) instrument (Rollins et al. 2020). The largest spike in
HNO$_3$ coincides with similar very large spikes in both NO and NO$_2$, indicative of both high total NOx and active photolysis.
However, there are periods during the night in which HNO$_3$ is generally elevated and also episodically peaks in concentration.
This is likely indicative of nighttime production pathways, presumably hydrolysis of N$_2$O$_5$ with H$_2$O on aerosol surfaces
and/or NO$_3$ reactions with dimethyl-sulfide (DMS) or volatile organic compounds (VOCs) (Brown et al. 2004; Reed et al.
2017). Representative mass spectra showing peak fits for both HOBr and HNO$_3$ as measured with the iodide CIMS are shown
in figure A3. These in-situ observations demonstrate the capability of the t-IMR to not only be deployed to the field, but also to
measure a diverse suite of compounds down to sub-ppt concentrations while minimizing wall effects and instrument residence
time.

## 6    Conclusions

In this work we have characterized a novel atmospheric-pressure IMR inlet (t-IMR) for use with CIMS applications. The
crossflow t-IMR is shown to operate under laminar flow conditions with an electric field used to accelerate reagent ions and
subsequent ion-ambient molecule clusters across the relatively high-pressure, high flow space to be sampled by the CIMS,
resulting in much higher observed ion signals compared to measurements with no applied field, typically by two orders of
magnitude. The standard operating electric potential is optimized to retain high total ion signal while also maximizing mea-
sured sensitivity. The t-IMR design is also shown to reduce wall memory effects compared to previously studied low-pressure
designs, decreasing instrument delay times from the order of minutes to the order of one second for a representative suite of
varied volatility organic gases. This increases the measurement reliability of ambient gases even at relatively lower volatilities,
particularly improving detection of low concentration species. Results from the first field deployment of the t-IMR demonstrate
the capability of the design to sample ambient trace gases continuously, down to sub-ppt concentrations in the case of HOBr.

Operating with the atmospheric pressure t-IMR shows several clear advantages, but also introduces some limitations, in-
cluding less control over ambient pressure and humidity changes. While the high flow through the IMR chamber reduces wall
interactions and residence time, this does present potential difficulty in background determinations as overflowing with 10 or
more L/min of UHP nitrogen gas can be difficult in certain settings where specialty gas quantities are limited. The relatively
high sample flow rate also makes it difficult to humidify such a "zero" in applications where a non-dry zero is desired, i.e.
with a water bubbler. Since CIMS sensitivity to many compounds has been shown to be somewhat dependent on the amount of
water vapor present in the IMR, it is important to quantify the humidity-dependence of sensitivities measured with the t-IMR



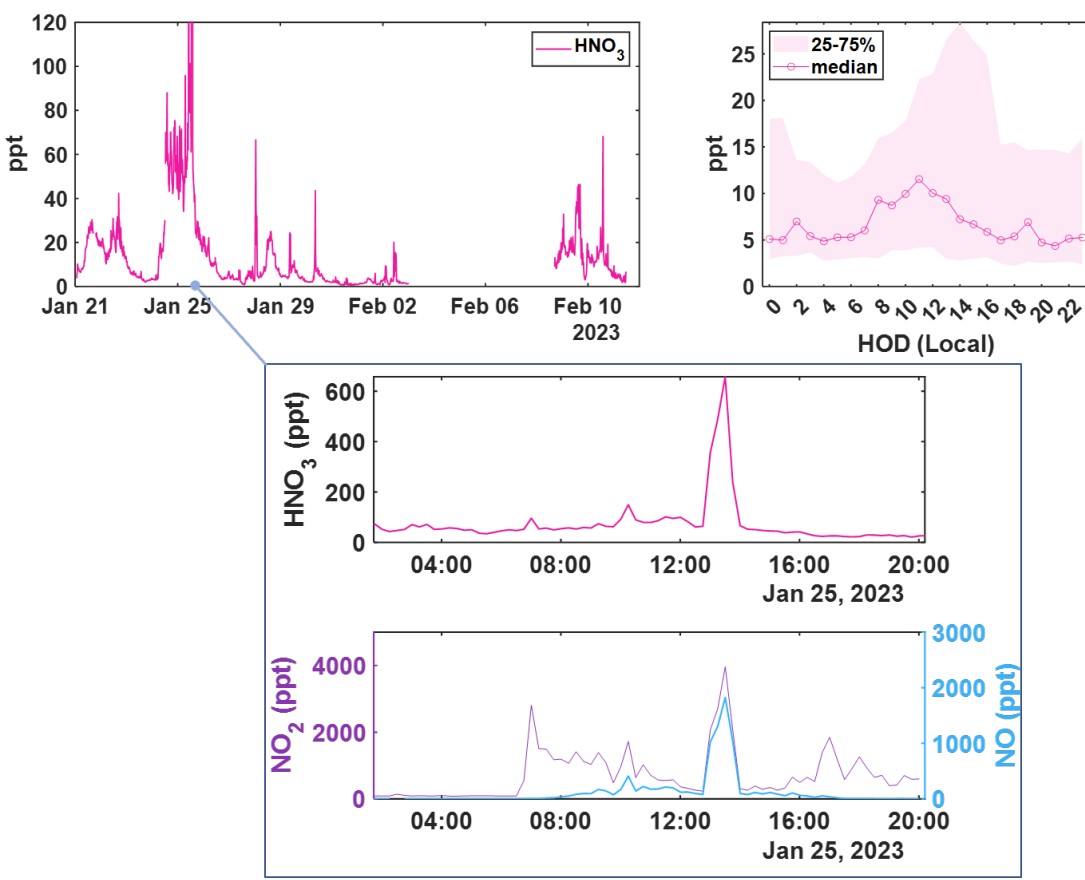

**Figure 5.** (top left:) 15-minute averaged timeseries of ambient $HNO_3$ concentrations measured during the winter BLEACH campaign and (top right) the diurnal profile showing mean signal at each hour of day (HOD) in local time. Shading indicates the bounds of the $25^{th}$ and $75^{th}$ quartiles for the measured concentrations in each hourly bin. (middle:) A zoomed in view of a spike in $HNO_3$ concentration as measured by CIMS coinciding with (bottom) measurements of NO (blue, right axis) and $NO_2$ (purple, left axis) as detected by a Laser-Induced Fluorescence (LIF) instrument



design. It would also be useful in the future to conduct a sensitivity study for the multitude of organic species that have been measured by iodide reagent chemistry with other IMR designs and conditions for inter-comparison purposes.





*Data availability.* IMR characterization data is available upon request to the authors.

**Appendix A**

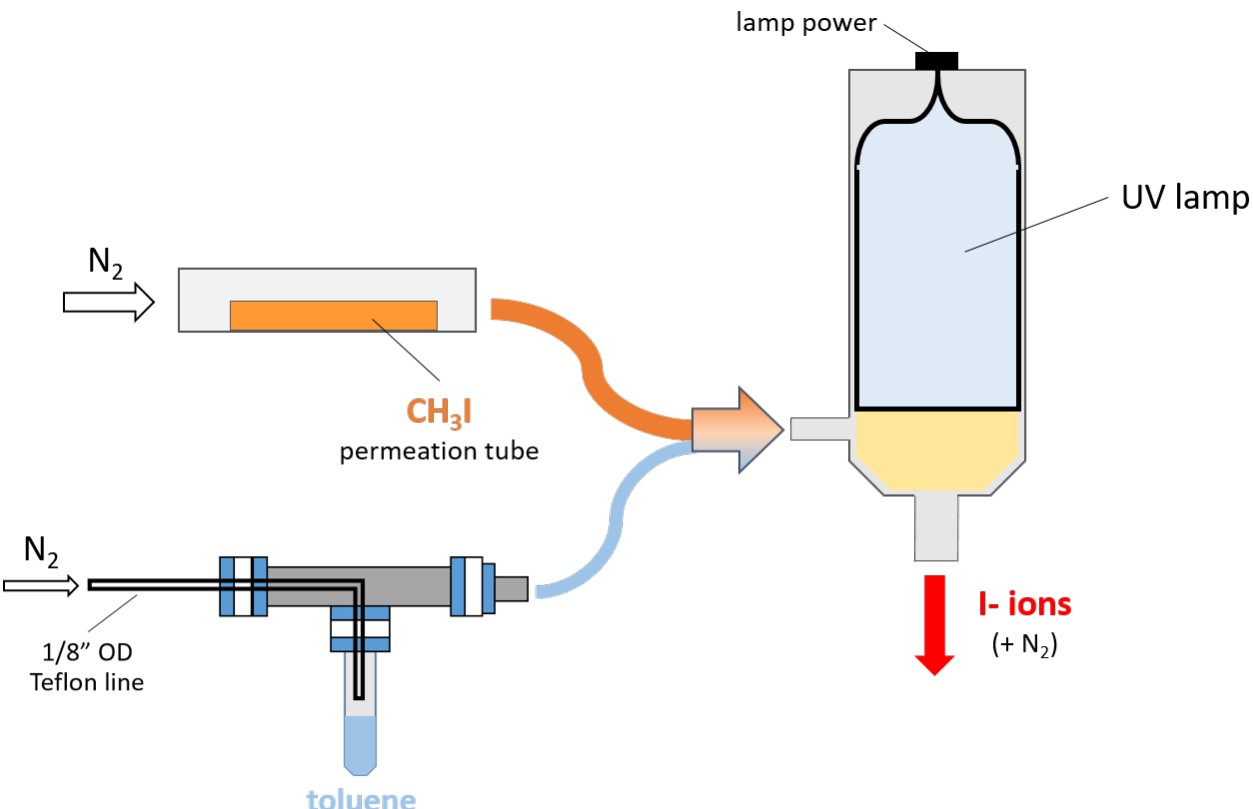

**Figure A1.** A diagram showing the VUV setup for I⁻ ion production. UHP $N_2$ is flown over both a $CH_3I$ permeation tube at 1.5 LPM (orange) and the headspace of a liquid toluene vial at 2 sccm (0.002 LPM, light blue), which combine and enter the VUV cavity illuminated by the UV lamp represented with yellow shading. Resulting I⁻ ions exit the VUV housing at a stainless 1/4" OD port where an electric potential is applied.





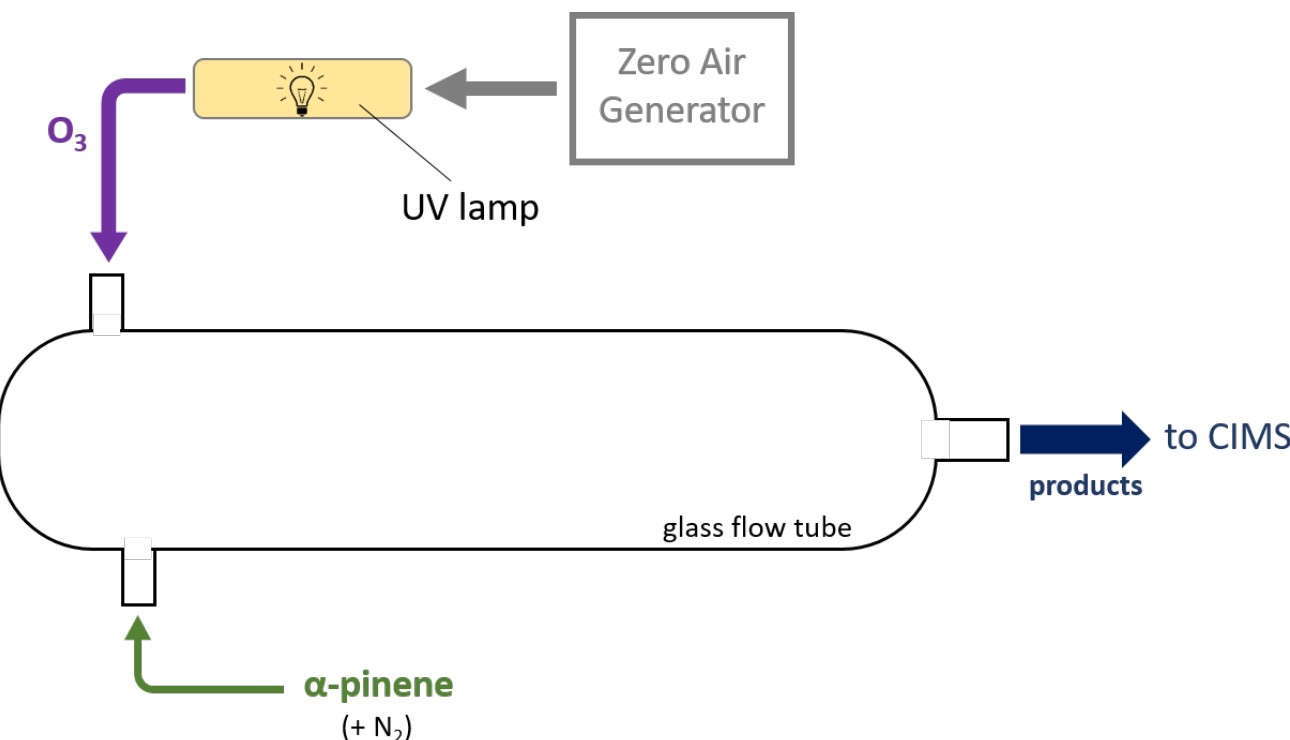

**Figure A2.** Setup schematic for the alpha-pinene ozonolysis wall effects experiment. A 4 LPM flow from the zero air generator is passed through a chamber illuminated by UV light to produce $O_3$ which is input into a glass flow tube (purple) along with a 200 sccm (0.200 LPM) flow of gaseous alpha pinene + UHP $N_2$ (green). Gases mix to form products (dark blue) which are output into the transverse IMR-equipped CIMS.



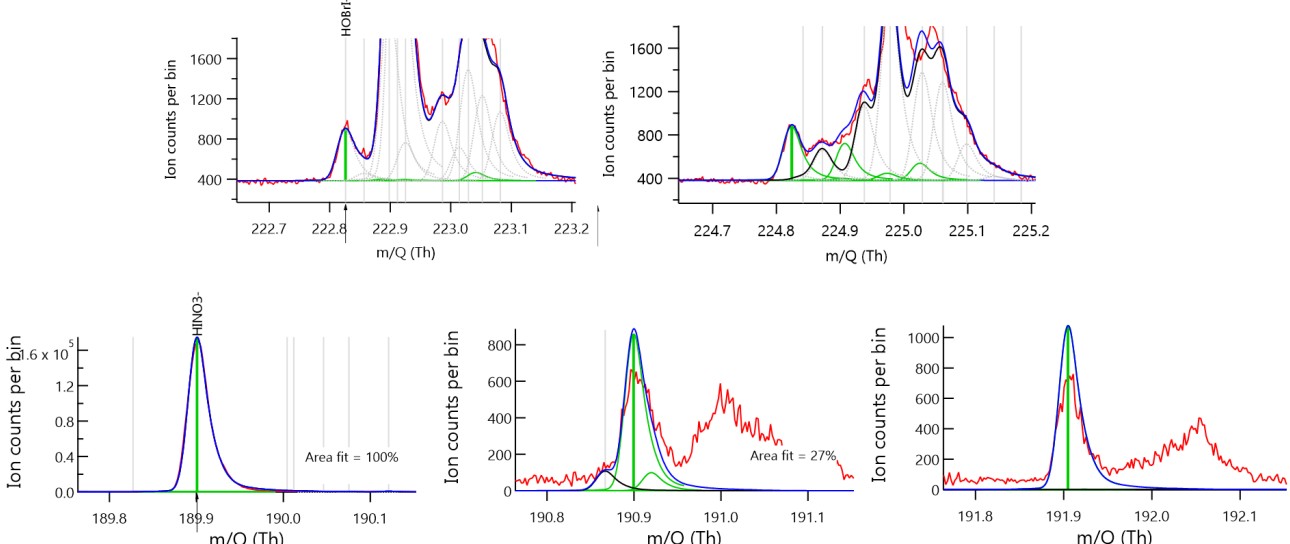

**Figure A3.** Approximately 15-minute integrated mass spectra showing signal in ion counts vs. mass to charge ratio (m/Q). Red traces indicate raw data while the dark blue show fitted and assigned peaks using an adjusted Gaussian model. Green bars indicate the mass-to-charge positions and fitted intensities for chemical compositions (top) HOBr and (bottom) HNO$_3$ when clustered to I$^-$. The additional plots to the right show fits for the respective isotopes, m+2 for HOBr and both m+1 and m+2 for HNO$_3$.

*Author contributions.* PR wrote the manuscript. PR, BHL, and JAT designed, tested, and deployed the t-IMR design for in-situ observations. GN and JTV provided NO and NO$_2$ observational data, in addition to assisting with BLEACH deployment. SI provided binding enthalpy calculations. All authors contributed to revisions of the manuscript.

*Competing interests.* The authors declare that they have no conflict of interest.

*Acknowledgements.* We would like to give enormous thanks to machinist Dennis Canuelle at the University of Washington for his assistance in the design and production of the t-IMR. We also thank the CSC IT Center for Science in Espoo, Finland, for providing the computing resources and funding from the Research Council of Finland project 355966 for their contributions to binding enthalpy calculations used in this work. Work at the Tudor Hill Marine Atmospheric Observatory was made possible by NSF award OCE-2123053 at the time of in-situ data collection, and we thank our collaborators at the Bermuda Institute of Ocean Sciences, particularly Andrew Peters. This work is supported by the National Science Foundation grant number 2109323.





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
