# Peer review of "Laboratory and Field Characterization of an Atmospheric Pressure Transverse Chemical Ionization Ion-Molecule reaction Region"

_EGUsphere, 2025_

## Referee Comment (RC1)

Review for AMT-2025-3709, "Laboratory and field Characterization of an Atmospheric Pressure Transverse Chemical Ionization Ion-Molecule reaction Region" by Rund et al.

The authors describe an updated design of a transverse atmospheric pressure Ion Molecule reaction Region (t-IMR) for use with chemical ionization mass spectrometers (CIMS). The authors describe in detail the details of operating and optimizing this t-IMR and demonstrate the capability of measuring trace gases at a ground site in the marine boundary layer. The data presented in this manuscript is quite impressive for difficult to measure sticky trace gases such as HOBr and $HNO_3$, and should be published. I have concerns about how the instrument could be calibrated for analytes difficult to deliver in the field, as the IMR conditions are not held constant (i.e. water vapor, pressure) during deployment and would require chamber experiments at field conditions in order to measure sensitivity as a function of these conditions. I would like to see the data for $IH_2O/I$ included in this manuscript to make the operation conditions of this system clearer to other users.

**Detailed comments:**

Line 11, "Finally, we demonstrate the capability of the t-IMR…", it would be useful to report sensitivity and LODs for HOBr, $Br_2$ and $HNO_3$ in the abstract. It would be even better to compare these LODs and sensitivities to the standard 'low-pressure' IMR design.

Line 25, add $CF_3O^-$, to this list, one of the major adduct ion chemistries.

Line 31, maybe add the design impacts backgrounds as well, which doesn't impact sensitivity or selectivity directly, but is important in instrument performance, and one of the benefits of the t-IMR design.

Line 46, "And of equal importance…", I think the authors a referring to wall effects and effectively the importance of quantifying backgrounds in the instrument accurately? As worded it is a little unclear with the authors mean.

Line 88, "Teflon$^{TM}$…" is this FEP or PFA?

Line 100, "via a gaseous mix…" Please specify mixing ratios and flowrates of MeI and toluene

Line 112, "The orifice, highlighted in purple in the inset, is designed specifically to maintain the pressure difference between the IMR at a somewhat dynamic atmospheric pressure on the order of ~1000 mbar and SSQ operating at 2 mbar." Does the orifice change diameter? How does it maintain a pressure difference between the IMR and SSQ? Is the pressure in the tIMR monitored continuously?

Line 116, "…to provide a small concentric flow of UHP nitrogen…" how much flow is this, how is it controlled?

Line 166, "$d$ is the diameter of the IMR tube (m)." One could consider the characteristic length in the Re calculation as the IMR pipe length rather than the diameter. The authors should also calculate Re this way to ensure the system is laminar.

Line 197, "Here, between 12 and 14 LPM of nitrogen is introduced…" so does this mean backgrounds are determined dry? How do the authors correct for analyte sensitivity differences between dry and wet IMR conditions? How long and often are backgrounds determined?

Line 203, "Low pressures…" This sentence and the following seem out of place in the paragraph.

Line 223, "$Br_2I^-$ cluster peak…", which one?

Line 229, "However, maximizing kinetic energy…" There is this effect but could the increase in field strength also simply be de-clustering the product clusters?

Line 258, "To characterize our t-IMR…" this section should be moved to a separate section in the Methods.

Line 285, "…especially after accounting for the t-IMR residence time of 0.75s" Could you include the Palm et al. low pressure IMR residence time as a comparison point here? My understanding is this was on the order of 50 - 100 ms?

Line 286, "This result is also comparable…" did the authors also do this experiment for $HNO_3$? It would a nice comparison to the Zhao et al. 2017 paper result.

Line 290, "The t-IMR was deployed…" much of this section could be moved to the Methods section as a "Field Deployment" section.

Line 306, "LOD is also calculated…" Please include a table of these LODs and for what integration time.

Line 315, "binding enthalpies…" Please move these DFT methods to a separate methods section.

Line 326, where did this equation come from? Was a water dependence of the Br2 or HOBr sensitivity considered?

Line 330, "The daily maximum levels of HOBr…" It would be useful to include Br2 and BrO data in this figure and analysis to confirm this measurement. Additionally, isotopologue ratios is normal practice when reporting halogens and should be included in the SI.

Line 336, "However, in this case, the signals are converted to atmospheric concentrations using a direct calibration of the t-IMR after the campaign." It is not clear why this was done rather than using the same approach as HOBr. Please show the equivalent result with the BE scaling approach. What was the sensitivity for HNO3? How was the water dependence of HNO3 sensitivity determined and applied?

Line 351, "However, there are periods during the night…" The iodide CIMS measures N2O5 and HPMTF very sensitively, if either of these chemistries are at play the authors have data to prove this. Please include this or remove the speculative statement.

---

## Author Comment (AC1)

Here we copy over the comments of Reviewer 1 for our preprint, with our responses to each in red bullet points. We greatly appreciate the discussion and points raised.

The authors describe an updated design of a transverse atmospheric pressure lon Molecule reaction Region (t-IMR) for use with chemical ionization mass spectrometers (CIMS). The authors describe in detail the details of operating and optimizing this t-IMR and demonstrate the capability of measuring trace gases at a ground site in the marine boundary layer. The data presented in this manuscript is quite impressive for difficult to measure sticky trace gases such as HOBr and HNO3, and should be published. I have concerns about how the instrument could be calibrated for analytes difficult to deliver in the field, as the IMR conditions are not held constant (i.e. water vapor, pressure) during deployment and would require chamber experiments at field conditions in order to measure sensitivity as a function of these conditions. I would like to see the data for IH2O/I included in this manuscript to make the operation conditions of this system clearer to other users.

 We have added a figure to the appendix showing representative changes in Iand IH2O- ion signals during the BLEACH23 campaign. We agree that by not holding IMR conditions (i.e. humidity) constant, this creates the need for corrections and laboratory tests to characterize sensitivity behavior against various conditions for species of interest.

**Detailed comments:**

Line 11, "Finally, we demonstrate the capability of the t-IMR...", it would be useful to report sensitivity and LODs for HOBr, Br2 and HNO3 in the abstract. It would be even better to compare these LODs and sensitivities to the standard 'low-pressure' IMR design.

• We have added a table with LOD and sensitivity values for key compounds

Line 25, add CF3O-, to this list, one of the major adduct ion chemistries.

Added the mention

Line 31, maybe add the design impacts backgrounds as well, which doesn't impact sensitivity or selectivity directly, but is important in instrument performance, and one of the benefits of the tIMR design.

Added the mention of background signals to this statement

Line 46, "And of equal importance...", I think the authors a referring to wall effects and effectively the importance of quantifying backgrounds in the instrument accurately? As worded it is a little unclear with the authors mean.

We combined sentences and changed wording here to be more clear. What we
mean is that the goal is to disturb and change analyte molecules as little as
possible to ensure measurements effectively capture ambient composition, which
does include wall effects.

Line 88, "TeflonTM..." is this FEP or PFA?

PFA, and we now say so in the manuscript

Line 100, "via a gaseous mix..." Please specify mixing ratios and flowrates of Mel and toluene

 We do not have exact mixing ratios measured to report, but we have moved up the statement about flow rates and conditions under which reagent flows are generated

Line 112, "The orifice, highlighted in purple in the inset, is designed specifically to maintain the pressure difference between the IMR at a somewhat dynamic atmospheric pressure on the order of ~1000 mbar and SSQ operating at 2 mbar." Does the orifice change diameter? How does it maintain a pressure difference between the IMR and SSQ? Is the pressure in the tIMR monitored continuously?

 We've changed the wording and specified in the text that the orifice diameter does not change, but was chosen to keep the SSQ pressure near 2 mbar when the IMR operates at atmospheric pressures (on the order of 1000 mbar)

Line 116, "...to provide a small concentric flow of UHP nitrogen..." how much flow is this, how is it controlled?

 We have specified the flow rate of 200 sccm (via an Alicat mass flow controller) in the text

Line 166, "d is the diameter of the IMR tube (m)." One could consider the characteristic length in the Re calculation as the IMR pipe length rather than the diameter. The authors should also calculate Re this way to ensure the system is laminar.

The IMR pipe length is much larger than the diameter, when using this value as
the characteristic length instead, the reynolds number reduces to Re = 24. So we
take the larger of the two in the text, which is still well under the values of 2000
required for a laminar flow regime.

Line 197, "Here, between 12 and 14 LPM of nitrogen is introduced..." so does this mean backgrounds are determined dry? How do the authors correct for analyte sensitivity differences between dry and wet IMR conditions? How long and often are backgrounds determined?

 Background determinations are dry, backgrounds were attempted almost twice daily in the field using an automatic delivery system which proved to be problematic, so 5 manual robust background periods are used for BLEACH data. Water vapor dependent sensitivities are applied to the key compounds for which we have experimentally determined relationships (see below comment and new appendix section on PH2O dependence)

Line 203, "Low pressures..." This sentence and the following seem out of place in the paragraph.

We agree, and moved this section

Line 223, "Br2I- cluster peak...", which one?

• m/Q 284.74, constrained isotopically at the m+2 and m+4 values, now mentioned in the manuscript

Line 229, "However, maximizing kinetic energy..." There is this effect but could the increase in field strength also simply be de-clustering the product clusters?

That is also possible, and is now included in the text

Line 258, "To characterize our t-IMR..." this section should be moved to a separate section in the Methods.

 To preserve the logical flow of the paper from IMR description to the individual demonstrations of abilities and improvements, we've chosen to leave the brief details of experimental setups at the beginning of each section

Line 285, "...especially after accounting for the t-IMR residence time of 0.75s" Could you include the Palm et al. low pressure IMR residence time as a comparison point here? My understanding is this was on the order of 50 - 100 ms?

• What was meant here is that the 1-2 second delay is made even less significant when subtracting the residence time of 0.75 s, not that our residence time is shorter than Palm's, which appears to be 100 ms based on a brief mention in their paper. We've deleted the phrase involving 0.75 s to avoid confusion.

Line 286, "This result is also comparable..." did the authors also do this experiment for HNO3? It would a nice comparison to the Zhao et al. 2017 paper result.

• We have performed this experiment after submission and added the results to the appendix (and a mention in the corresponding section)

Line 290, "The t-IMR was deployed..." much of this section could be moved to the Methods section as a "Field Deployment" section.

See previous comment re: flow of article and methods

Line 306, "LOD is also calculated..." Please include a table of these LODs and for what integration time.

Now included in a table with LOD and sensitivity

Line 315, "binding enthalpies..." Please move these DFT methods to a separate methods section.

• See previous comment re: flow of article and methods

Line 326, where did this equation come from? Was a water dependence of the Br2 or HOBr sensitivity considered?

 An explanation is added for the equation and experiments were conducted on water dependence after manuscript prep, results have now been added here.
 We've added an appendix (B) showing a water-vapor dependent curve of Br2 and HNO3 sensitivity based on both in-field calibrations and laboratory experiments. The HOBr and HNO3 timeseries and diurnal profiles have been updated to reflect this dynamic sensitivity, which does not change the behavior of either significantly, nor the conclusions of this work.

Line 330, "The daily maximum levels of HOBr..." It would be useful to include Br2 and BrO data in this figure and analysis to confirm this measurement. Additionally, isotopologue ratios is normal practice when reporting halogens and should be included in the SI.

 The high resolution spectra are shown in the appendix to confirm the isotope ratio of the identified HOBr peak fit. Br2 and BrO data are included a forthcoming comprehensive Br observations manuscript. We aim here to demonstrate the deployment ability of the t-IMR and low LODs, and would like to save potential discussion of observed bromine chemistry.

Line 336, "However, in this case, the signals are converted to atmospheric concentrations using a direct calibration of the t-IMR after the campaign." It is not clear why this was done rather than using the same approach as HOBr. Please show the equivalent result with the BE scaling approach. What was the sensitivity for HNO3? How was the water dependence of HNO3 sensitivity determined and applied?

 We now use the BE method for HNO3 sensitivity and have applied the above mentioned water vapor correction Line 351, "However, there are periods during the night..." The iodide CIMS measures N2O5 and HPMTF very sensitively, if either of these chemistries are at play the authors have data to prove this. Please include this or remove the speculative statement.

 We have now added and referred to a figure in the appendix showing N2O5 as well as CINO2 signals increasing with HNO3 and NO2 overnight to support this statement

---

## Author Comment (AC2)

Here we copy over the comments of Reviewer 2 for our preprint, with our responses to each in blue bullet points. We greatly appreciate the discussion and points raised.

**Major Comments**

- 1. Since the t-IMR represents an improvement over traditional IMR designs, it would be valuable to provide additional quantitative data. For example, the authors could briefly summarize the limits of detection, sensitivity, and wall effects for representative species under conventional configurations. This would allow readers to more directly appreciate the extent of the improvements achieved with the new IMR.
  - we have added a table summarizing LODs and sensitivities for key compounds, in addition to adding results of an HNO3 wall effects experiment to the appendix
- 2. The authors acknowledge that one limitation of the t-IMR is the inability to control humidity. This raises the question of how the reported concentration data were calculated during field observations. Was it assumed that humidity has no effect on sensitivity? Given that specific concentration values are presented and that these species are known to be influenced by humidity in traditional IMRs, a detailed description of how humidity effects on sensitivity were accounted for is essential. Furthermore, was any on-site calibration performed during the observation period? If so, these data could be used to further analyze and discuss the role of humidity. Since the treatment of humidity directly determines whether the instrument is truly field-deployable, and given that the authors have already applied the t-IMR in field studies, it would be highly valuable to provide recommendations regarding the handling of humidity in such applications.
  - We have added and referred to a new section in the appendix outlining the water vapor dependence of Br2 and HNO3 sensitivity as we were able to test it in the lab. Time series figures now reflect a dynamic humidity-dependent sensitivity used to convert signals to concentrations.
- 3. Overly descriptive text Several sections (e.g., Sect. 2, description of dimensions and housing) read like a technical manual. While detail is important for reproducibility, the writing could be more concise and structured (perhaps moving some details to the Appendix).
  - We've removed some tertiary dimensions and technicalities that are not necessary for the demonstrations and conclusions of the manuscript
- 4. Axial vs. transverse geometry (discussion point) Many atmospheric-pressure CIMS instruments (e.g., HOx-CIMS) employ an axial geometry in which the sampled flow is directed straight into the pinhole. By contrast, this study adopts transverse geometry. It

would be useful for the authors to briefly comment on the rationale behind this choice and how it may compare in terms of sensitivity, turbulence, or wall effects. This would help contextualize the work for readers familiar with axial designs.

 A few sentences have been added in the IMR description section about the choice of transverse geometry preventing neutral compounds from diluting and contaminating analyte flows, as well as reducing potential for turbulence and eddy formation at the capillary MS entrance. We also report our reynolds number and compare wall effects directly with Palm et al's coaxial design.

**Minor Comments**

L216: I understand that it may be challenging to evaluate the dependence of sensitivity on humidity in the laboratory given the high sampling flow rate. However, was the potential variation of sensitivity under different humidity conditions examined during field calibrations? In traditional IMRs, for example, Br2 is known to be significantly affected by humidity within certain ranges. How was the influence of humidity accounted for during the BLEACH observations?

 We now include and reference an appendix section on water vapor dependent sensitivities calculated and applied to bromine and nitric acid measurements.
Corresponding figures have been updated

L306: The authors should report explicit limits of detection. Although the calculation formula may be cited, the authors should provide a worked example using Br2 that lists all parameter values used in the computation. Because the LOD is a key criterion for assessing whether the t-IMR outperforms traditional IMRs, presenting the full set of inputs and the resulting LOD for Br2 is essential for a transparent and fair comparison.

• We have added the parameters for Br2 in a statement where we reference Bertram's equation.

L328: The reported Br2 sensitivity of 1.5 counts ppt-1 per 1e6 TRIC is substantially lower than the value shown in Figure 2. Please clarify the source of this discrepancy. If the value of 1.5 counts ppt-1 per 1e6 TRIC is correct, it appears lower than that of traditional IMRs. Please discuss whether the t-IMR design prioritizes a lower LOD at the expense of sensitivity, and explain the underlying factors that could account for this trade-off.

 The experiment performed for figure 2 was in a laboratory environment while the 1.5 is reported for the in situ measurements at THMAO, which is much more humid and has higher temperatures, affecting sensitivity. Different CIMS tuning voltages also likely contribute to this discrepancy, as a significant amount of time

- exists between this experiment and BLEACH, with tuning adjustments performed in the field.
- It does seem that compared to low pressure designs, at least at such high humidity, the t-IMR sacrifices sensitivity for lower LOD, likely stemming from ion-molecule interaction time, large sample flows, and reduced wall effects/background signals
- L341: The specific sensitivity of HNO3 should be reported. If the background concentration of HNO3 cannot be subtracted, it is unclear why this compound was nevertheless selected for demonstration. I suggest that the authors briefly explain the rationale for choosing HOBr and HNO3 as representative species in their analysis.
  - HNO3 was selected because it is well known to be less volatile and deposit on IMR surfaces contributing to wall effects. The inability to be background subtracted is only due to one fitting in the N2 flow delivery apparatus for zeroing, likely contaminated from an experiment with HNO3 performed before BLEACH. HNO3 is shown because, despite not being background subtracted, the lowest concentrations recorded (assumed to be larger than or equivalent to instrument background or noise) are shown to be relatively small, with a minimum of 0.3 ppt and a 10th percentile value of 1 ppt. HNO3 sensitivity is now reported in a table with that of the other key species

**Other Comments**

The authors are advised to carefully check the manuscript for grammatical issues as well as the correct formatting of superscripts and subscripts. Below, I list only the instances I have identified.

 The following formatting and grammar issues have all been resolved in the manuscript

L3: "shows significant improvements in potential measurement interference" it is awkward to say improvements in interference. Mitigation or reduction sounds better.

L4: "reduce", without "s".

L7:"... exhibited by alpha-pinene ozonolysis" sounds ungrammatical. "...as demonstrated in α-pinene ozonolysis experiments" sounds better.

L16: "affects" instead of "effects".

L38:" However, by reducing the pressure of the IMR dilutes sample molecular number concentration by up to several orders of magnitude." Delete "by".

L41: Change "For already low concentration specie" to "For species present at very low concentrations".

L82: "time series" instead of "timeseries", in here and the rest of the manuscript.

L159: "its" instead of "it's"

L200: The "2" in N2 should be formatted as a subscript (N2).

L231: "their" instead of "them"

L238: "factor of 2" instead of "factor 2".

L254 "from" instead of "off of".

L293: "... located on the southwestern shores of Bermuda operated by the Bermuda Institute of Ocean Sciences (BIOS)." lack of conjunction "and".